# Development of Spirulina-Enriched Fruit and Vegetable Juices: Nutritional Enhancement, Antioxidant Potential, and Sensory Challenges

**DOI:** 10.3390/foods14203539

**Published:** 2025-10-17

**Authors:** Biljana Cvetković, Miona Belović, Lato Pezo, Jasmina Lazarević, Goran Radivojević, Mirjana Penić, Olivera Šimurina, Aleksandra Bajić

**Affiliations:** 1Institute of Food Technology, University of Novi Sad, Bulevar cara Lazara 1, 21000 Novi Sad, Serbia; miona.belovic@fins.uns.ac.rs (M.B.); jasmina.lazarevic@fins.uns.ac.rs (J.L.); olivera.simurina@fins.uns.ac.rs (O.Š.); aleksandra.bajic@fins.uns.ac.rs (A.B.); 2Institute of General and Physical Chemistry, Studentski trg 12/V, 11000 Belgrade, Serbia; latopezo@yahoo.co.uk; 3Faculty of Sciences, University of Novi Sad, Trg Dositeja Obradovića 3, 21000 Novi Sad, Serbia; goran.radivojevic@dgt.uns.ac.rs (G.R.); mirjana.penic@dgt.uns.ac.rs (M.P.)

**Keywords:** antioxidant activity, functional beverages, nutritional enhancement, sensory analysis, sour cherry juice, spirulina, texture, tomato juice

## Abstract

Spirulina (*Arthrospira platensis*) is a protein- and antioxidant-rich microalga, but its use in beverages is limited by sensory acceptance. Four juices (apple, sour cherry, tomato, and celery) were initially tested with added blue and green spirulina (0.8% and 1.6% *w*/*w*). Based on preliminary acceptability scores, only sour cherry and tomato juices were selected for further analyses. Blue spirulina enrichment increased protein (from 0.80 to 1.36 g/100 g in sour cherry; 0.89 to 1.52 g/100 g in tomato), fat (0.05 to 0.21 g/100 g; 0.09 to 0.25 g/100 g), and energy (259 to 279 kJ/100 g; 140 to 170 kJ/100 g). Antioxidant activity improved significantly, with DPPH IC_50_ reduced from 260 to 135 mg/mL (sour cherry) and from 268 to 171 mg/mL (tomato). Colour analysis confirmed a shift from red to blue hues (a* from 15.2 to 3.7 in sour cherry). Data were statistically processed using ANOVA followed by Tukey’s HSD test, while sensory data were additionally evaluated by PCA and GPA to identify product-specific differences. These results confirm that spirulina-enriched sour cherry and tomato juices are nutritionally enhanced functional beverages, though sensory off-notes remain a challenge for consumer acceptance.

## 1. Introduction

The global demand for innovative food products with improved nutritional profiles and potential bioactive effects has been steadily increasing over the past decade [1,2]. The consumption of such products is significant in the prevention of major non-communicable diseases such as obesity, diabetes, and cardiovascular disorders, while also promoting healthier dietary habits [3,4,5]. Among alternative sources of functional food ingredients, microalgae are considered especially promising due to their naturally encapsulated bioactive compounds that provide important health benefits [6,7]. Moreover, microalgae represent a newly recognized protein resource with substantial global exploitation potential [8,9]. When considering microalgae as the “food of the future,” their incorporation into food products must overcome technological and sensory limitations [10,11,12]. It is therefore essential to define incorporation thresholds that preserve product structure and sensory characteristics, particularly colour and taste [13,14]. Among microalgae, *Arthrospira platensis* (spirulina) is the most widely studied because of its high nutritional value and health-promoting properties, including antioxidant, immunomodulatory, and anti-inflammatory activities. Spirulina contains both enzymatic and non-enzymatic antioxidants, vitamins C, E, K, and B-complex, β-carotene, provitamin A, chlorophyll, and phycocyanin, as well as 18 amino acids and essential minerals such as Ca, Mg, Fe, Cu, Se, and Zn [15]. Clinical studies suggest that phycocyanin is the principal antioxidant compound, while also serving as a natural blue pigment in foods [15,16]. Despite these benefits, consumer acceptance is limited by spirulina’s distinctive odour and taste, especially in primary commercial forms such as tablets or capsules [15]. Current research trends, therefore, focus on incorporating spirulina into food matrices that improve sensory acceptance while enhancing nutritional value [15]. Previous studies have investigated its addition to various juices, such as apple [15,17], pomegranate [18], grape [19], and date nectar [20], either as powder or extract. To identify suitable juice matrices for spirulina enrichment, four widely consumed juices—apple, sour cherry, tomato, and celery—were initially screened. Sour cherry and tomato juices showed the highest acceptability and were selected for further evaluation. Sour cherry juice (*Prunus cerasus* L.) is rich in anthocyanins and ascorbic acid, linked to antioxidant, anti-inflammatory, and antidiabetic effects [21,22]. Tomato juice is valued for its lycopene and other bioactive compounds with established health benefits. In this context, the present study aimed to evaluate spirulina-enriched juices using a reverse approach compared to the standard procedure. Instead of starting from predefined juice matrices, four juices were screened for sensory acceptability. Based on this evaluation, sour cherry and tomato juices were selected as the most promising candidates. Subsequent analyses focused on the effects of blue spirulina enrichment on their chemical composition, antioxidant activity, Colour, texture, and sensory characteristics, to assess their potential as functional beverages. In this study, the focus was placed on evaluating the overall functional response of fruit and vegetable juices enriched with spirulina—through antioxidant potential, colour, texture, and sensory properties—as integrative indicators of bioactive compound activity. The detailed quantification of individual vitamins, pigments, and minerals was beyond the scope of this research but represents a planned direction for future investigation. Furthermore, this study introduces a reverse, sensory-based selection framework that supports the technological optimization of spirulina-enriched beverages. Such an approach bridges sensory evaluation with product formulation, contributing to the advancement of processing strategies and practical applications in the development of functional drinks with improved stability and consumer appeal. Moreover, Spirulina enrichment combines a potent source of natural antioxidants with fruit and vegetable juices that serve as convenient carriers for innovative, health-oriented products, thereby expanding the functional beverage market and meeting the needs of modern consumers.

## 2. Materials and Methods

### 2.1. Preparation of Juices

Four basic juices were selected based on their popularity and health benefits: two fruit juices (apple and sour cherry) and two vegetable juices (tomato and celery). The fruits and vegetables used in this study were common Serbian varieties: Idared apple, Oblačinska sour cherry, stalk celery (*Apium graveolens* var. *dulce*), and processing-type tomato (“šljivar” cultivar). These fruits and vegetables were purchased from a local market, and the juices were extracted using a Bosch MES 3500 culinary juicer. To each juice, green spirulina powder was added at two levels: 2 g (0.8% *w*/*w*) and 4 g (1.6% *w*/*w*) per 250 mL serving. The same supplementation scheme was applied using blue spirulina powder, with additions of 2 g and 4 g per 250 mL portion. In total, twenty juice samples were prepared for the initial acceptability test using both blue and green spirulina powder. Homogenization of juice samples after spirulina addition was performed using an Ultra-Turrax T25 digital homogenizer (IKA, Staufen, Germany) operating at 10,000–13,000 rpm for 1–2 min to ensure uniform dispersion of the powder in the liquid matrix. Blue and green Spirulina powders were purchased from NatureHub (Belgrade, Serbia; http://www.naturehub.rs, accessed on 10 July 2025). According to the manufacturer’s declaration, the blue spirulina powder consisted predominantly of phycocyanin extracted from *Arthrospira platensis*, while the green spirulina powder represented the dried biomass of *Arthrospira platensis*. The composition of the blue spirulina powder used in this study was as follows: dry matter 84.00%, moisture 14.00%, protein 40.00%, fat 0.32%, ash 10.54%, total sugars 1.00%, calculated carbohydrates 33.14%, sodium 1100 mg/100 g, salt content 0.30%, with an estimated energy value of 1255.22 kJ (295.44 kcal) per 100 g. These values correspond to the manufacturer’s declaration and are consistent with literature data for phycocyanin rich *Arthrospira platensis* powder.

### 2.2. Basic Chemical Composition

The dry matter, ash, protein, and fat content were determined using methods from the Association of Official Analytical Chemists (AOAC, 2000): specifically, methods 925.10, 925.51, 950.36, and 935.38 [23]. Total sugars were measured using the Schoorl method (AACC method 80-68, 2000) [24]. Total carbohydrates and energy values were calculated based on the analyses. Salt content was estimated from the sodium content determined by atomic absorption spectrometry according to ISO 6869:2008 [25]. In brief, the samples were incinerated in a muffle furnace at a temperature of 550 ± 15 °C. The resulting ash was dissolved in hydrochloric acid (HCl). The diluted sample was then atomized in an air–acetylene flame, and the metal content was quantified using an atomic absorption spectrophotometer, specifically the SpectrAA-10 (Varian, Belrose, Australia). All chemical analyses were conducted in triplicate.

### 2.3. Total Phenolic Content and Antioxidant Activity

The total phenolic content (TPC) was determined using the method described by Singleton, Orthofer, and Lamuela-Raventós (1999) [26], adapted for a plate reader (Multiskan Ascent, Thermo Electron Corporation, Waltham, MA, USA). To begin, 125 μL of 0.1 M Folin–Ciocalteu reagent was added to 25 μL of the extracts. The mixture was allowed to react for 10 min before adding 100 μL of a 7.5% *w*/*v* sodium carbonate solution. After a 2 h incubation period, the absorbance was measured at 690 nm. To account for any interference, a correction was made by replacing the reagents with an equal volume of distilled water. A standard curve was prepared using Gallic acid, and the TPC was expressed as milligrams of Gallic acid equivalents (GAE) per gram of juice dry weight. All experiments were performed in triplicate. The determination of free radical scavenging activity was assessed by monitoring the transformation of the 2,2-diphenyl-1-picrylhydrazyl (DPPH) radical in the presence of antioxidants, following the method described by Espín, Soler-Rivas, and Wichers (2000) with slight modification [27]. The reaction mixture in each well contained 10 μL of the sample, 60 μL of DPPH solution, and 180 μL of methanol. The control consisted solely of methanol, while the correction included 10 μL of the sample and 240 μL of methanol. After incubating the mixtures in the dark at room temperature for 60 min, the absorbance was measured at 492 nm using a plate reader. The DPPH radical scavenging capacity was calculated using the following equation:% DPPH radical scavenging capacity = 100 − ((Asample − Acorrection) × 100)/Acontrol

The IC_50_ value indicated the concentration required to scavenge 50% of the DPPH radicals. Each sample was tested at five different concentrations, and all experiments were performed in triplicate.

The Ferric Reducing Antioxidant Power (FRAP) test was conducted following a modified procedure from Benzie and Strain (1999) [28]. The FRAP reagent was prepared by mixing 300 mM acetate buffer (pH 3.6), 10 mM 2,4,6-tris(2-pyridyl)-s-triazine (TPTZ) dissolved in 40 mM hydrochloric acid, and 20 mM ferric chloride in a volume ratio of 10:1:1 (*v*:*v*:*v*). In a 96-well plate, 10 µL of the sample, 225 µL of the FRAP reagent, and 22.5 µL of distilled water were combined. For the control, the sample was replaced with an equal volume of distilled water, while the correction contained distilled water instead of the FRAP reagent. Absorbance was measured after 6 min at 630 nm. Ascorbic acid was used to create a standard curve, and the results were expressed as milligrams of ascorbic acid equivalents (AAE) per gram of juice dry weight. Each analysis was performed in triplicate.

### 2.4. Colour and Texture Measurements

The colour of juice samples was measured using a Chroma Meter CR-400 (Konica Minolta Co., Ltd., Osaka, Japan) equipped with a CM-A98 attachment designed for liquid samples. The measurements were conducted in triplicate, using a D65 light source and an observer angle of 2°. The parameters measured included CIE L* (lightness), CIE a* (where +a* indicates redness and −a* indicates greenness), CIE b* (where +b* indicates yellowness and −b* indicates blueness), CIE C* (chroma), and the hue angle (h°). Prior to taking measurements, the tristimulus values from the CIE Lab* readings were calibrated against a standard white plate (Y = 84.8; x = 0.3199; y = 0.3377).

Texture analysis was performed using a TA.XT Plus Texture Analyzer (Stable Micro Systems, Godalming, UK). The consistency of the juice samples was determined using the Back Extrusion Rig (A/BE), which was equipped with a 45 mm disc and an extension bar, utilizing a 5 kg load cell. Tests were conducted in a standard back extrusion container (50 mm diameter), filled to approximately 65%, and the samples were tempered for 30 min at room temperature (23 ± 2 °C). Instrumental settings were taken from the sample project YOG1_BEC within the Texture Exponent Software (TEE32, version 6.1.6.0, Stable Micro Systems, Godalming, UK) and were modified to set a distance of 20 mm and a trigger force of 5 g. All measurements were conducted in triplicate.

### 2.5. Sensory Analysis of Juices

#### 2.5.1. Initial Acceptability Test

The initial selection of juices for the experiments was determined through a sensory acceptability test. Eight trained panellists—six females and two males, aged 30 to 50 years and employed at the Faculty of Sciences or the Institute of Food Technology at the University of Novi Sad—evaluated the juices based on their overall acceptability using a seven-point hedonic scale (from 1 = extremely dislike to 7 = extremely like). The juice samples were assigned random three-digit codes, and the tests were conducted under controlled sensory laboratory conditions [29]. Juices that received an acceptability score higher than 4 on the hedonic scale (as shown in Appendix A) were included for further experimentation, along with their respective control (baseline) juices. Six juices were selected for further study: sour cherry juice (SC1) with the addition of 2 g (SC2) and 4 g (SC4) of blue spirulina per portion, and tomato juice (T1) with the addition of 2 g (T2) and 4 g (T4) of blue spirulina per portion. Although the initial set of juices included apple, sour cherry, tomato, and celery, apple and celery were excluded from further analysis due to their low acceptability scores, and no nutritional analyses were performed on those juices.

#### 2.5.2. Descriptive Analysis

The same eight trained panellists who conducted the initial acceptance testing were brought together again to perform a sensory profiling of juices through descriptive analysis. All panellists had over two years of experience in examining both commercial products and products developed in research projects focused on fruits and vegetables. Their prior training included exercises on identifying sensory descriptors, developing terminology, and evaluating the intensity of sensory attributes. For this sensory evaluation, a two-hour session was conducted to finalize the list of descriptors. Initially, the panellists were provided with terms from previously published studies, but they were encouraged to keep, add, or remove any terms as needed [30,31]. A final list of descriptors was established through a consensus approach, and definitions can be found in Appendix A. Intensity assessment used an unstructured linear scale, ranging from 0 (not perceptible) to 100 (very intense) [32]. The juice samples were coded with random three-digit numbers and presented to the panellists simultaneously. To cleanse the palate between individual juice samples, they were provided with water that had low mineral content and unsalted crackers. The assessment took place in a sensory testing laboratory where environmental conditions were adequately controlled [29].

### 2.6. Statistical Analysis

Statistical analysis of the data was performed using XLSTAT 2018.7. The sensory results are presented as mean values ± standard deviation (SD). To evaluate the homogeneity of variances, Levene’s test was used, and the Shapiro–Wilk test was applied to assess normality. Analysis of variance (ANOVA), followed by Tukey’s HSD post hoc test, was conducted to identify sensory attributes that significantly distinguished between the samples and to evaluate variations in the sensory profiles of the juice samples. The descriptive sensory data were also analyzed using Principal Component Analysis (PCA), which helped create a sensory map to visualize the positions of assessors in relation to overall liking scores [33]. Correlations among colour, sensory, and texture parameters—considering both sensory and instrumental measurements—were examined using R software version 4.0.3 (64-bit). Additionally, a further PCA was conducted to differentiate the juice samples, utilizing Statistica version 14.0.0.15 [34].

## 3. Results and Discussion

### 3.1. Basic Nutritional Composition, Total Phenolic Content and Antioxidant Activity

Since apple and celery juices were not selected after the preliminary acceptability test, nutritional and further analyses were conducted only for sour cherry and tomato juices. The addition of blue spirulina powder increased the dry matter, ash, protein, fat, total carbohydrates, total sugar, salt, and energy value of both the tomato juice and sour cherry juice, as expected based on previous research [35,36]. However, there were no statistically significant changes in the total phenolic content (Table 1 and Table 2). Conversely, the antioxidant activity of both juices increased with the addition of blue spirulina powder, with a more pronounced effect observed in the DPPH test compared to the FRAP test [37,38]. Previous studies have reported that antioxidant blends of fruit and vegetable juices, particularly those rich in polyphenols, carotenoids, and vitamin C, can exert synergistic effects on oxidative stress reduction and overall health improvement [39,40,41]. The present study builds upon these findings by exploring spirulina fortification as a novel strategy for enhancing antioxidant activity through the addition of natural pigments and bioactive proteins. Unlike conventional formulations, this research integrates a sensory-based selection of juice matrices prior to compositional and functional assessment, thereby establishing a more application-oriented framework for developing functional beverages

### 3.2. Colour and Texture Measurements

A noticeable change in the colour of the fruit juice was observed after adding spirulina in various proportions (see Figure 1). This change was further supported by instrumental analysis of the colour parameters. The addition of blue spirulina powder decreased the lightness (L*) and colour saturation (C*) of both cloudy sour cherry juice and tomato juice. Simultaneously, the intensity of the red tone (a*), typical of sour cherry and tomato juices, also diminished. In the case of sour cherry juice, the yellow tone (b* > 0) shifted to blue (b* < 0), which is reflected in the changes to the colour angle (h°). Increasing the amount of spirulina powder enhanced the blue tone, indicated by lower b* values and colour angle values approaching 270°. The observed colour changes in sour cherry juice were primarily due to pigment blending between native anthocyanins and the phycocyanin–chlorophyll pigments present in spirulina. No visible degradation of anthocyanins was detected, and the hue shift reflected additive colour mixing rather than pigment instability. The addition of spirulina slightly increased the pH of sour cherry juice, consistent with the mildly alkaline nature of the spirulina powder, but this change was insufficient to affect anthocyanin stability. For tomato juice, the intensity of the yellow tone (b* > 0) decreased with higher spirulina content, nearly reaching zero in sample T4. Additionally, one study identified correlations between the C-phycocyanin content of spirulina and colour parameters, noting a positive correlation with a* values and a negative correlation with b* values [35].

The textural properties of cloudy sour cherry juice—specifically firmness, consistency, cohesiveness, and viscosity index—were not significantly affected by the addition of blue spirulina powder. This indicates that there were no differences between the control and the enriched samples (see Table 3). In contrast, the addition of spirulina powder did significantly alter the textural properties of tomato juice. We observed lower firmness and consistency, while cohesiveness and the viscosity index were higher, with a statistically significant difference noted in the viscosity index. However, only one study directly addresses the incorporation of spirulina. Fradique et al. investigated the effects of Spirulina maxima in pasta and found that higher concentrations of biomass (ranging from 0.5% to 2.0%) led to increased firmness [42].

The available evidence indicates that interactions between particles significantly influence the viscosity of tomato juice. Research has shown that the interaction between pulp particles plays a crucial role in determining viscosity [43]. Key factors affecting rheological parameters include particle concentration, size, and morphology [44]. Firmness is defined as the peak force during probe penetration, while consistency refers to the area under the curve up to that peak. It was hypothesized that mixing spirulina powder into tomato juice disrupts the pulp network, leading to reduced firmness and consistency [45]. Conversely, the back-extrusion test revealed that the enriched juices exhibited a higher cohesiveness and viscosity index, likely due to the increased number of dispersed particles and the friction among them. However, since the cohesiveness values did not differ significantly from the control, existing filling and packaging equipment can be utilized without modifications. Both enriched sour cherry and tomato juices can be packaged in standard formats for cloudy juices, with the usual recommendation to shake before consumption.

### 3.3. Sensory Evaluation

The addition of microalgae transformed both juices from dark red to dark blue, although turbidity remained unchanged. Panel assessments indicated an increase in viscosity at concentrations of 0.8% and 1.6% blue spirulina (see Table 4). The discrepancy between panel-rated viscosity and instrumental viscosity is attributed to differing shear histories in shear-thinning systems, characterized by rapid probe withdrawal versus slow swirling [45]. Granularity increased in both juice matrices, but the change was significant only for the tomato juice (see Table 4). In terms of flavor, the enrichment with microalgae reduced the characteristic sour-cherry/tomato odor and taste while enhancing algae notes in a dose-dependent manner. Although sweetness tended to decrease, this change was not statistically significant. As highlighted in the previous literature, the addition of spirulina often produces distinct off-notes, necessitating flavor masking techniques [46]. Nonetheless, the nutritional and bioactive benefits of microalgae are well established, underscoring their potential role in functional food development despite sensory challenges [47]. Our findings further support this, as higher concentrations of spirulina led to a marked increase in algae-like odor and flavor. Additionally, the tomato juice became significantly less salty, which is consistent with mineral-related taste interactions, and there was a slight but significant increase in bitterness at the 1.6% concentration [48]. Sourness increased with dosage in both juices but was significant only for the tomato juice (see Table 4).

The sensory evaluation results reveal notable variation among the assessed attributes. Turbidity recorded the highest mean score (96.22 ± 12.35), indicating that this attribute was consistently perceived with high intensity across all samples, accompanied by relatively low variability. Both characteristic flavour (74.56 ± 25.50) and sour taste (66.67 ± 25.00) also exhibited high average intensities but with greater dispersion, suggesting more heterogeneous perceptions among assessors. The blue colour presented a broad scoring range (0–100) and a high standard deviation (47.66), reflecting significant differences in visual assessments between samples. Moderate mean values with moderate variability were observed for viscosity (59.33 ± 14.61), characteristic odour (59.81 ± 31.59), and salty taste (25.36 ± 27.70). In contrast, lower mean scores were recorded for granularity (21.36 ± 14.08), algae odour (22.81 ± 24.32), algae flavour (14.56 ± 17.54), and sweet taste (16.86 ± 8.84), suggesting that these attributes were generally perceived at low to moderate intensities. Attributes related to off-notes, such as mould odour (0.17 ± 0.70), mould flavour (0.86 ± 2.18), and bitter taste (0.33 ± 0.99), demonstrated very low mean values and minimal variability, indicating that these undesirable sensory properties were largely absent in the evaluated samples.

Additionally, a panel analysis was conducted using a Mixed Models approach with Type III Sum of Squares to evaluate the significance of various sources of variation, including products, assessors, and their interactions. ANOVA was applied to all dependent variables according to the specified model:(1)Y=μ+P+A+P×A

In this equation, P represents the product (a fixed factor), while A (assessor) and P × A are random factors. The results of the panel analysis are summarized in Table 5.

For each descriptor, we calculated the Type III sum of squares from the ANOVA of the selected model. The panel analysis, presented in Table 5, indicated significant differences among the products. The analysis of variance revealed that several sensory descriptors showed a statistically significant product effect (*p* < 0.05), meaning these attributes can effectively differentiate between the evaluated products. Significant product effects were observed for blue colour, viscosity, granularity, characteristic odor, algae odor, characteristic flavor, algae flavor, salty taste, and bitter taste, with blue colour demonstrating the strongest discrimination (*F* = 24,663.07, *p* < 0.0001). In contrast, turbidity, mold odor, mold flavor, sweet taste, and sour taste did not exhibit significant product-related differences (*p* > 0.05). Consequently, these parameters were excluded from further analysis due to their lack of statistically significant influence. Assessor effects were also statistically significant (*p* < 0.05) for most descriptors, including turbidity, viscosity, granularity, characteristic odor, algae odor, characteristic flavor, mold flavor, sweet taste, salty taste, and sour taste. This suggests variability in scoring patterns among the panelists. Notably, no significant product × assessor interactions were detected, indicating that assessors ranked the products consistently, without major differences in perception trends. The results confirm that visual, textural, odor, and taste descriptors—particularly blue colour; viscosity, and salty taste—are the most effective for differentiating the products in the tested sample set. The sensory evaluation results across six products (SC1, SC2, SC4, T1, T2, and T4) and six trained assessors show clear differences in perceived intensity scores (Table 6). Products T2 and T4 generally received higher mean scores compared to T1, especially from assessors such as A1, A2, and A5, indicating a greater perceived intensity for the evaluated attributes in these samples. The SC-series products exhibited greater variability; SC2 consistently garnered high scores (e.g., 18.39 from A1 and 16.77 from A5), while SC1 received low ratings across all assessors (scores mostly ranged between 2.27 and 3.63). While assessor scoring trends were generally consistent in ranking high- and low-intensity samples, individual differences in scoring amplitude were notable. For instance, A3 provided a markedly low score for T1 (2.20) but the highest score for T4 (15.26), indicating a greater contrast in perception. Overall, the results suggest that product type strongly influences sensory scores, with T2, T4, and SC2 standing out as the most intense, SC1 as the least intense, and T1 displaying moderate to low ratings. These patterns indicate a statistically relevant product effect, which can be confirmed through ANOVA or multivariate sensory profiling.

### 3.4. Generalized Procrustes Analysis (GPA) and Principal Component Analysis (PCA)

GPA (Generalized Procrustes Analysis) is recognized as an effective multivariate method for aligning assessor configurations and minimizing scale effects [49]. This method was utilized to reduce scale effects and obtain a consensus configuration, which allows for the comparison of terminology and evaluation patterns among assessors. The resulting consensus matrix represents the mean product scores across all assessors, with residuals indicating deviations from this consensus—high residuals signify greater divergence, while low residuals indicate stronger agreement. Principal Component Analysis (PCA) of the consensus matrix identified the main sources of variation, with Factors *F1* through *F5* explaining the largest proportions of variance. The combination of product scores, assessor residuals, and eigenvalues provides a comprehensive view of both consensus and disagreement in sensory evaluations. Results from the Panel Analysis of Variance (PANOVA) indicate that, within the Generalized Procrustes Analysis framework, translation effects were the only source of variation with a statistically significant contribution (*F* = 2.913, *p* < 0.001). This suggests that there were notable positional adjustments among assessor configurations to align with the consensus space. In contrast, scaling effects (*F* = 1.614, *p* = 0.179) and rotation effects (*F* = 0.520, *p* = 0.998) were not significant, indicating minimal differences in the magnitude and orientation of assessor configurations. Similar tests of translation, scaling, and rotation effects have been described in statistical approaches to GPA [50]. The relatively high residual sum of squares after scaling (5665.27) and after rotation (6808.29) reflects the remaining variation not explained by these adjustments. Residuals by product (Figure 2a) reveal that SC4 exhibited the largest deviation from the consensus configuration (1309.05), followed by SC2 (1103.52) and T2 (900.71), indicating greater disagreement among assessors regarding these products. Conversely, T1 showed the lowest residual (697.83), suggesting higher agreement among panel members in its evaluation. These findings confirm that while positional adjustments among assessor spaces were essential, the panel generally maintained consistency in scaling and orientation, with major differences arising from specific perceptions of each sample. Additionally, the residuals by assessor (Figure 2b) illustrate the degree of deviation between each assessor’s individual configuration and the consensus configuration derived from Generalized Procrustes Analysis. Lower residual values indicate stronger alignment with the panel consensus, while higher values reflect greater divergence and potential assessor-specific biases or variations in scoring patterns.

Among the panel, assessors A4 (343.75) and A3 (426.38) showed the closest agreement with the consensus, indicating consistent and representative evaluations. Assessor A2 (612.72) also demonstrated good alignment, though with slightly higher variability. In contrast, assessors A6 (1792.97) and A5 (1380.56) exhibited the largest deviations, reflecting substantial differences in their scoring relative to the panel average. Additionally, assessor A1 (1108.90) showed above-average divergence. These results reveal that while most assessors were reasonably consistent, a few displayed notable departures from the consensus, which could influence panel reproducibility and should be considered in training or calibration sessions. The scaling factors reflect variations in the range or intensity of scores used by each assessor compared to the consensus configuration in the Generalized Procrustes Analysis. A scaling factor close to 1.000 indicates that the assessor used a scoring range similar to the panel average. Values greater than 1.000 suggest a tendency to use a wider scoring range (indicating higher dispersion), while values below 1.000 indicate a more restricted use of the scale. In this panel (Figure 3), assessors A3 (1.139) and A2 (1.112) applied the broadest scoring ranges, suggesting they differentiated more between the samples in their evaluations. Assessor A1 (0.971) was close to the consensus range, showing balanced scale usage. Assessors A4 (0.945), A5 (0.940), and A6 (0.945) had the lowest scaling factors, indicating a more conservative use of the scoring scale. The variation in scaling factors is relatively small, implying that assessors generally maintained similar scoring ranges, with only minor differences in the intensity of scale usage.

The results of the consensus test indicate a strong agreement among assessors in the sensory evaluation. The consensus coefficient (*Rc* = 0.8846) is close to 1, reflecting a high level of concordance among the panel’s assessments. The number of permutations (300) represents the iterations used to evaluate the statistical significance of the consensus, while the quantile value (100.0) confirms that the observed consensus is at the maximum percentile of the permutation distribution. Overall, these results demonstrate that the panel provided consistent and reliable evaluations, supporting the robustness of the sensory data for subsequent analysis. The dimensions test results indicate that all five factors (*F1–F5*) extracted from the sensory data are statistically significant in explaining the variability among the products. The observed *F*-values for each factor are as follows: *F1* = 496.07, *F2* = 105.42, *F3* = 32.04, *F4* = 23.34, and *F5* = 11.41. These values are substantially higher than the corresponding critical *F*-value (Critical = 2.534), and the associated *p*-values (<0.0001) confirm that these differences are highly significant at α = 0.05. The quantile values for *F1–F4* (100.0) indicate that the observed *F*-values exceed all estimates based on permutations, highlighting the robustness of these dimensions in capturing sensory variation. Although *F5* has a slightly lower quantile (67.33), it remains statistically significant. These results demonstrate that all five factors contribute meaningfully to differentiating the products, with *F1* accounting for the largest proportion of variance, followed sequentially by *F2*, *F3*, *F4*, and *F5*. The analysis confirms the existence of multiple independent dimensions underlying the panel’s sensory evaluations. Furthermore, the extraction of higher-order dimensions beyond the first two aligns with recent advances in multivariate approaches to sensory analysis [51].

The PCA biplot (Figure 4), based on the GPA consensus configuration, provides a clear visualization of the sensory relationships among juice samples. The application of PCA to analyze consensus matrices and explore sample differentiation has been well-documented in sensory profiling [52,53]. The first two principal components (PC1 and PC2) explain a significant proportion of the total variability, with PC1 accounting for 77.97% and PC2 for 16.84%, together capturing 94.81% of the total variance. This indicates that these two dimensions effectively summarize the main differences among the samples. Analysis of the eigenvectors reveals that PC1 is strongly associated with visual and flavor attributes: blue colour (−0.990), characteristic odour (0.997), characteristic flavour (0.988), and algae odour (−0.914). This suggests that these descriptors are the primary drivers of variance along PC1. On the other hand, PC2 is primarily influenced by salty taste (0.949), granularity (0.736), and viscosity (0.545), highlighting textural and taste-related differences among the samples. Other descriptors, such as bitter taste and algae flavor, contribute moderately to the higher-order components (*F3*–*F5*) and thus play a smaller role in differentiating the products. Examining the object coordinates, T1 and SC1 cluster on the negative side of PC1, indicating high levels of blue colour and algae odour but lower characteristic odor and flavor scores. Conversely, T2, T4, SC2, and SC4 are positioned on the positive side of PC1, reflecting higher intensities of characteristic odor and flavor. Along PC2, T4 and T2 are positioned positively, suggesting higher levels of granularity and viscosity, while SC2 and SC4 are on the negative side, indicating lower scores for these attributes. Overall, the PCA confirms that visual and flavor attributes dominate PC1, while textural and taste attributes are mainly captured by PC2. This allows for a clear differentiation of the juice samples based on their sensory profiles. A comparable PCA-based discrimination of microalgae-enriched food products has been demonstrated, where visual and textural changes were key discriminating factors [54].

According to the correlation analysis of the products evaluated by assessors during the sensory evaluation of juices (Figure 5), a clear separation was observed for sample T4. In contrast, the boundaries between samples T1 and SC1, as well as between SC2 and SC4, were less distinct. The coordinates obtained for each assessor across different configurations indicate a high degree of consistency in the sensory structure, particularly along the first principal factor (*F1*). Across all assessors (A1–A6), T1 and SC1 consistently show large positive *F1* scores, which indicate that they share similar sensory profiles characterized by the same key attributes. Conversely, T2, T4, SC2, and SC4 consistently display large negative *F1* scores, confirming their differentiation from T1 and SC1 along the primary sensory dimension. The second factor (*F2*) exhibits greater variability among assessors, reflecting differences in how secondary attributes were weighted. For instance, T4 often achieves high positive *F2* values (e.g., 34.59 for A2, 39.34 for A6), while V2 and V4 consistently occupy the negative *F2* space. This suggests that the texture or taste-related descriptors associated with *F2* were perceived differently by these groups. Higher-order factors (*F3–F5*) have smaller magnitudes and greater variability, indicating that these dimensions explain less variance and are more sensitive to individual assessor perceptions or noise. Notably, some samples (e.g., V4 for assessors A5 and A6) exhibit strong loadings on *F3* and *F5*, which may reflect specific, less dominant sensory traits perceived by certain assessors. Overall, the coordinate patterns confirm that the panel reached a stable consensus in distinguishing the primary sensory differences (*F1*), with moderate variability in the secondary dimensions (*F2*–*F5*) attributable to individual differences in sensory weighting.

### 3.5. Correlation Analysis and Principal Component Analysis

A Colour Correlation Analysis (CCA) was performed to investigate the similarities in the chemical, textural, and sensory profiles of various products. The findings are illustrated graphically in Figure 6. The correlation analysis identified several strong and statistically significant relationships linking sensory attributes, colour coordinates, and chemical composition. The blue colour exhibited very strong negative correlations with characteristic odour (*r* = −0.982, *p* = 0.001) and characteristic flavour (*r* = −0.961, *p* = 0.002), as well as with all three CIELAB colour coordinates (L*, a*, and b*) with *r* values ranging from −0.888 to −0.976 (*p* < 0.02). Conversely, it displayed positive correlations with algae odour (*r* = 0.856, *p* = 0.029) and sugar content (*r* = 0.822, *p* = 0.045). Viscosity showed positive correlations with granularity (*r* = 0.940, *p* = 0.005), algae odour (*r* = 0.945, *p* = 0.004), algae flavour (*r* = 0.925, *p* = 0.008), and bitter taste (*r* = 0.820, *p* = 0.046). It was also positively correlated with key proximate components such as protein (*r* = 0.903, *p* = 0.014) and fat (*r* = 0.914, *p* = 0.011). Granularity itself had positive relationships with algae odour (*r* = 0.816, *p* = 0.048) and protein (*r* = 0.913, *p* = 0.011). Characteristic odour was inversely correlated with algae odour (*r* = −0.930, *p* = 0.007) but showed a strong positive correlation with algae flavour (*r* = 0.991, *p* < 0.001) and the colour coordinates L*, a*, and b* (*r* > 0.873, *p* < 0.024). It also had strong negative correlations with sugars (*r* = −0.873, *p* = 0.023) and total carbohydrates (*r* = −0.871, *p* = 0.024). Algae odour was negatively associated with characteristic flavour (*r* = −0.965, *p* = 0.002) but positively linked to algae flavour (*r* = 0.856, *p* = 0.030) and major chemical components like protein (*r* = 0.901, *p* = 0.014) and fat (*r* = 0.908, *p* = 0.012). Characteristic flavour correlated positively with algae flavour (*r* = 0.849, *p* = 0.033), bitter taste (*r* = 0.886, *p* = 0.019), and all CIELAB coordinates, while being inversely related to sugars and total carbohydrates (*r* ≈ −0.91, *p* ≤ 0.012). Algae flavour had strong positive correlations with bitter taste (*r* = 0.951, *p* = 0.004), protein (*r* = 0.956, *p* = 0.003), and fat (*r* = 0.936, *p* = 0.006). Salty taste displayed strong negative associations with blue colour (*r* = −0.976, *p* = 0.001), protein (*r* = −0.957, *p* = 0.003), and total carbohydrates (*r* = −0.957, *p* = 0.003). Among the colour coordinates, L*, a*, and b* were all strongly intercorrelated (*r* > 0.939, *p* < 0.006) and positively associated with fat content (*r* > 0.962, *p* < 0.002). Regarding proximate composition, dry matter (DM) was negatively related to sugars (*r* = −0.824, *p* = 0.044) but showed a very strong positive correlation with both protein (*r* = 0.993, *p* < 0.001) and total carbohydrates (*r* = 0.995, *p* < 0.001). Protein and total carbohydrates were nearly perfectly correlated (*r* = 0.995, *p* < 0.001), while ash content was inversely correlated with both (*r* ≈ −0.88, *p* < 0.022). Overall, these results demonstrate a clear interdependence between sensory characteristics, colour parameters, and chemical composition. Specific compositional traits—particularly those related to protein, carbohydrates, and pigments—are identified as key drivers of sensory perception. Similar strong correlations among sensory descriptors, structural changes, and compositional traits have also been reported in fruit systems, where modifications in pectin were shown to influence texture and softening [55].

The PCA results (Figure 7) demonstrate that the first two principal components (PC1 and PC2) collectively account for 89.61% of the total variance in the dataset. This indicates a significant reduction in dimensionality with minimal information loss. Specifically, PC1 has an eigenvalue of 13.13, representing 46.91% of the total variance, while PC2 has an eigenvalue of 11.96, explaining an additional 42.70% of the variance. The high cumulative variance captured by these two components suggests that they effectively represent the dataset’s variability, making them suitable for biplot visualization and aiding in the interpretation of key underlying patterns. The use of PCA for dimensionality reduction and the analysis of complex multivariate food datasets is well established and has been widely utilized to correlate compositional parameters with sensory attributes [56,57]. PCA loading matrix illustrates the contribution of each variable to the first two principal components, which collectively explain 89.61% of the total variance.

PC1 (46.91% variance) is strongly and positively associated with various attributes such as granularity (6.90), ash (7.37), viscosity (6.44), salt (6.24), algae odor (5.03), fat (4.14), algae flavor (4.02), protein (3.69), salty taste (3.27), and bitter taste (3.11). In contrast, it shows strong negative loadings for the index of viscosity (−5.32), total carbohydrates (−5.27), sugars (−5.21), dry matter (−4.46), cohesiveness (−4.14), energy value (−4.07 to −4.03), characteristic flavor (−3.51), and characteristic odor (−2.80). This indicates that PC1 primarily separates samples based on a compositional and textural gradient, where higher mineral content, saltiness, and viscosity oppose higher carbohydrate content, energy value, and cohesiveness. PC2 (42.70% variance) is characterized by very strong negative loadings from CIELAB colour parameters b (−7.79), L (−7.51), C (−7.16), and a (−6.68), along with firmness (−5.26), characteristic odor (−4.91), salty taste (−4.36), consistency (−4.30), and characteristic flavor (−4.27). The highest positive loadings are observed for blue colour (5.54), hue angle (3.78), energy value (3.80–3.84), protein (3.57), fat (3.05), algae odour (2.59), sugars (2.58), and total carbohydrates (2.53). This suggests that PC2 differentiates samples mainly based on colour and firmness, contrasting lighter, more intensely coloured, and firmer samples with those that are bluer, rich in protein and fat, and have higher perceived algae odor and sweetness. PC1 captures the trade-off between dominance of minerals, protein, and viscosity versus the carbohydrate, energy, and flavor profile. In contrast, PC2 represents a sensory, colour, and composition axis that distinguishes lighter, firmer products from darker, blue-toned, protein- and fat-rich ones. The distribution of variables in both components indicates that compositional parameters (ash, protein, fat, carbohydrates) and colour attributes are the primary drivers for sample separation in the PCA space. While spirulina enrichment has improved the nutritional and functional properties of sour cherry and tomato juices, challenges related to sensory acceptance remain. The overall research design followed a “reverse approach,” where initial sensory screening was applied to identify juice matrices with the highest potential for consumer acceptability before conducting compositional and functional analyses. This strategy ensured that subsequent evaluations focused on products with practical market potential, thereby reinforcing the translational relevance of the findings. Future research should focus on optimizing formulation strategies, such as combining spirulina with masking agents or flavor enhancers, to improve consumer acceptance while preserving nutritional benefits. However, as noted in other studies, incorporating microalgae often leads to sensory challenges, including intense colour shifts and off-flavor development, requiring formulation strategies to balance nutritional advantages with consumer appeal [54].

## 4. Conclusions

The addition of blue spirulina to sour cherry and tomato juices significantly enhanced their nutritional value by increasing protein, fat, and energy content, while also boosting antioxidant activity. Colour analyses showed a noticeable shift in hue from red to blue, particularly in sour cherry juice. In terms of texture, sour cherry juice experienced limited effects, but tomato juice showed significant changes, including reduced firmness and consistency, along with an increased viscosity index. Sensory evaluation indicated distinct differences in colour and flavor. The enrichment led to a decline in the characteristic fruit and vegetable notes and an increase in the algae odor and flavor. Despite these sensory challenges, the cohesiveness values and overall handling of the products suggest that current filling and packaging equipment can still be used effectively. Therefore, spirulina-enriched sour cherry and tomato juices can be regarded as nutritionally enhanced functional beverages suitable for standard packaging formats, as long as consumer acceptance issues related to the off-notes of algae are addressed. This outcome confirms the effectiveness of the “reverse approach” applied in this study, demonstrating that sensory-based screening can serve as a practical and efficient tool for selecting suitable matrices prior to detailed nutritional and functional evaluation. Integrating sensory validation at the early stage of product development strengthens the overall methodological concept and ensures better alignment with consumer expectations. Although the present study did not evaluate the storage stability of spirulina-fortified juices, this represents a limitation and an important direction for future research, particularly in the context of product shelf life and commercialization potential. Future research should focus on advanced formulation approaches, including ingredient combinations and processing innovations, to further enhance the sensory appeal, nutritional quality, and stability of spirulina-enriched fruit and vegetable juices. Future research should focus on advanced formulation approaches, including ingredient combinations and processing innovations, to further enhance the sensory appeal and stability of spirulina-enriched fruit and vegetable juices.

## Figures and Tables

**Figure 1 foods-14-03539-f001:**
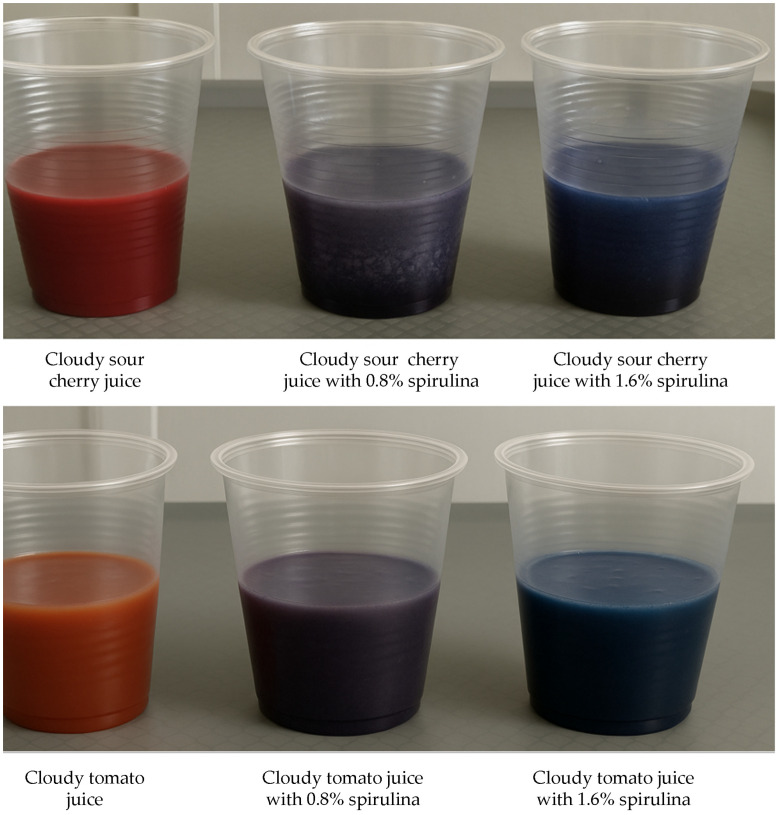
Sour cherry and tomato juice (SC1, T1) color change after adding 2 g (0.8%, SC2; T2) and 4 g (1.6%, SC4; T4) of spirulina per 250 mL portion.

**Figure 2 foods-14-03539-f002:**
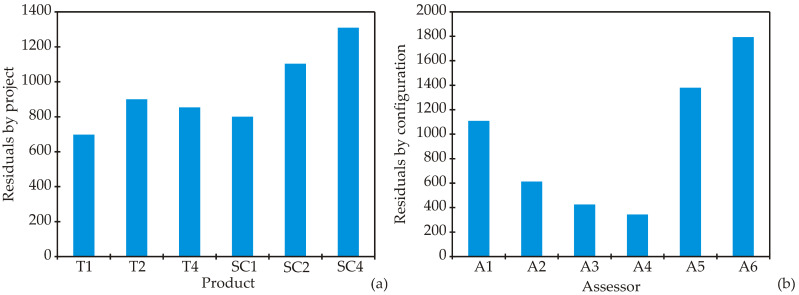
Residuals by (**a**) products and (**b**) assessors.

**Figure 3 foods-14-03539-f003:**
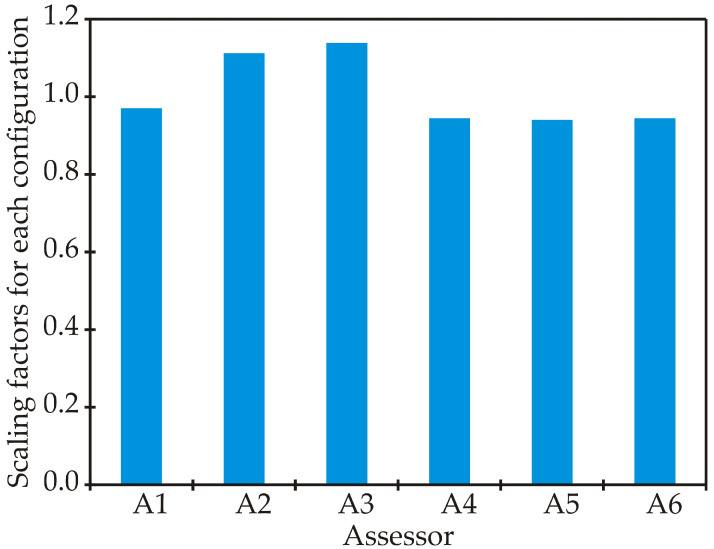
Scaling factors for Assessor’s scores.

**Figure 4 foods-14-03539-f004:**
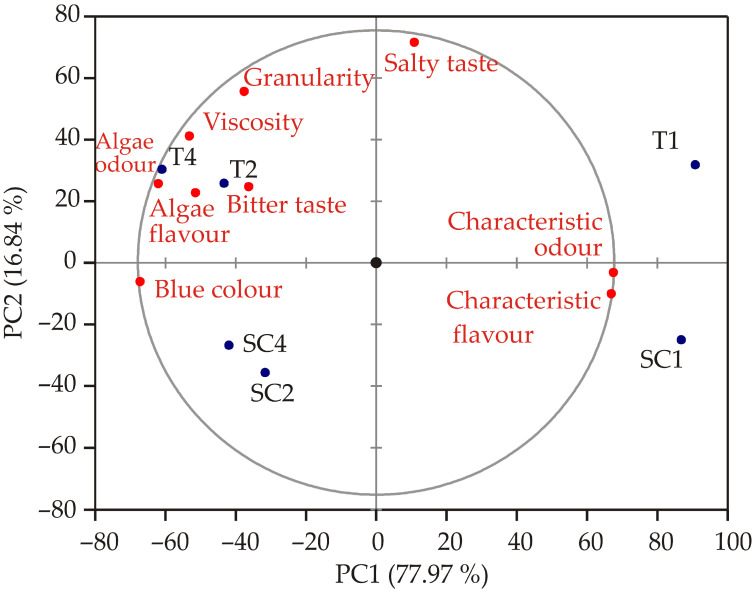
The PCA biplot diagram of the juice sensory analysis.

**Figure 5 foods-14-03539-f005:**
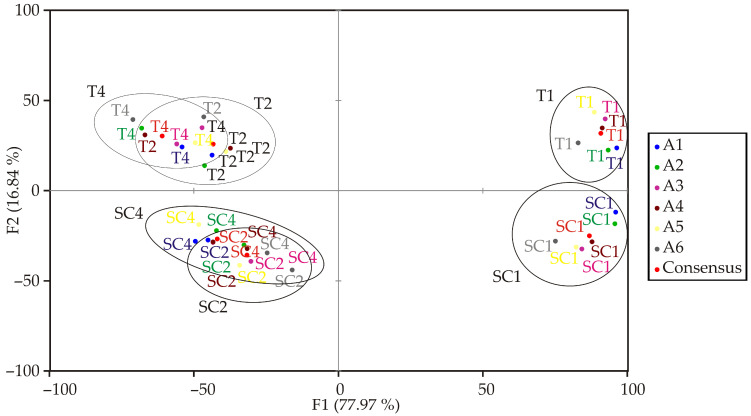
Correlation of products obtained by Assessors during sensory analysis.

**Figure 6 foods-14-03539-f006:**
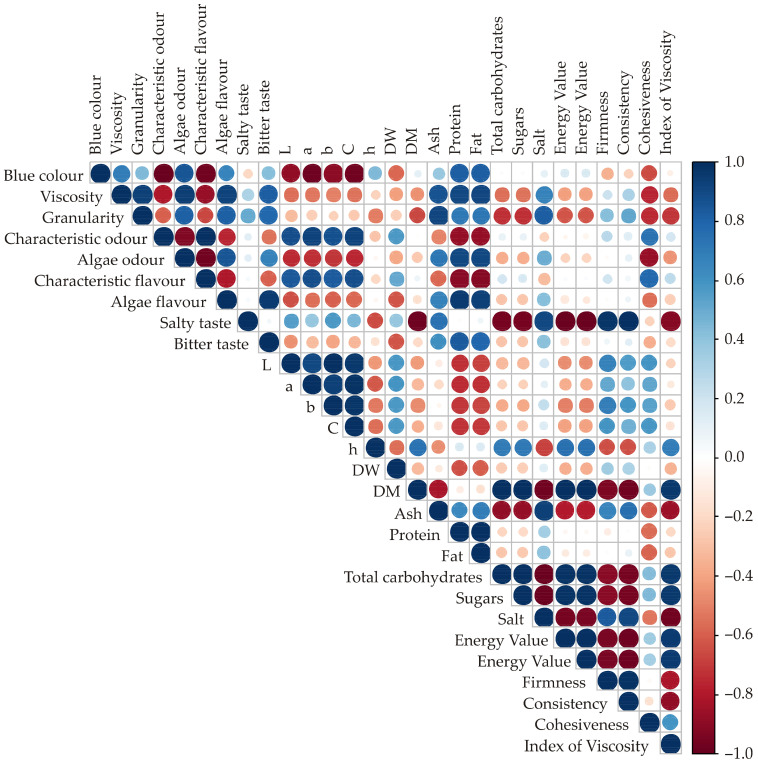
Colour correlation diagram between sensory analysis, colour, textural, and chemical parameters.

**Figure 7 foods-14-03539-f007:**
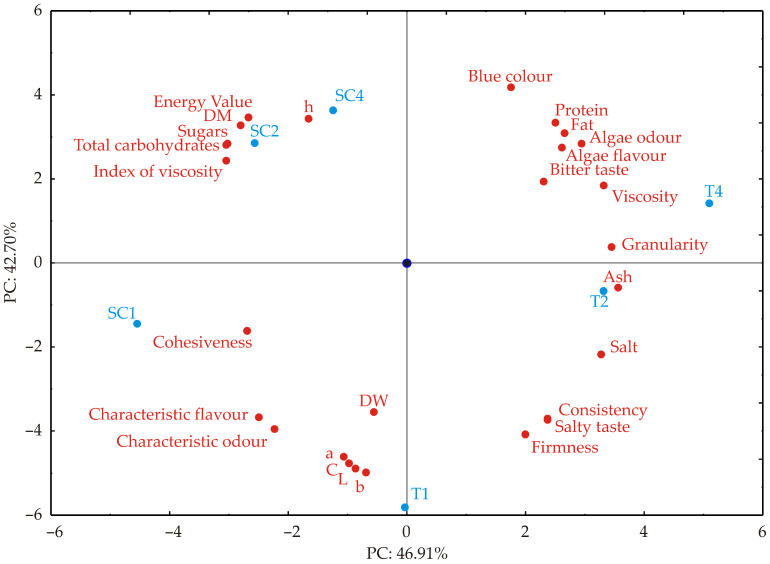
The PCA biplot diagram depicting the relationships among sensory analysis, colour, textural, and chemical parameters.

**Table 1 foods-14-03539-t001:** Basic nutritional composition (per fresh weight), total phenolic content and antioxidant activity of basic and enriched sour cherry juices.

Parameter	SC1	SC2	SC4
Dry matter (g/100 g)	15.21 ± 0.06 b	15.81 ± 0.13 a	15.97 ± 0.21 a
Ash (g/100 g)	0.90 ± 0.01 c	1.01 ± 0.03 b	1.13 ± 0.03 a
Protein (g/100 g)	0.80 ± 0.03 c	1.08 ± 0.03 b	1.36 ± 0.03 a
Fat (g/100 g)	0.05 ± 0.01 c	0.13 ± 0.01 b	0.21 ± 0.01 a
Total carbohydrates (g/100 g)	13.47 ± 0.04 a	13.59 ± 0.14 a	13.27 ± 0.23 a
Total sugars (g/100 g)	11.25 ± 0.05 b	11.30 ± 0.04 ab	11.38 ± 0.03 a
Salt (g/100 g)	0.05 ± 0.00 c	0.10 ± 0.00 b	0.15 ± 0.00 a
Energy value (kJ/100 g)	259.27 ± 1.77 b	273.20 ± 3.21 a	279.48 ± 4.65 a
TPC (mg GAE/g d.w.)	12.82 ± 0.30 a	12.88 ± 0.99 a	13.04 ± 1.74 a
DPPH IC_50_ (mg/mL)	260.19 ± 35.49 a	177.95 ± 46.97 b	134.79 ± 10.45 b
FRAP (mg AAE/g d.w.)	0.76 ± 0.08 a	0.99 ± 0.19 a	1.08 ± 0.27 a

Results are presented as mean ± standard deviation (*n* = 3). Results in a column marked with different letters are significantly different at the *p* < 0.05 level (Fischer’s LSD post hoc test).

**Table 2 foods-14-03539-t002:** Basic nutritional composition (per fresh weight), total phenolic content and antioxidant activity of basic and enriched tomato juices.

Parameter	T1	T2	T4
Dry matter (g/100 g)	8.45 ± 0.06 c	9.09 ± 0.13 b	9.73 ± 0.21 a
Ash (g/100 g)	1.27 ± 0.01 c	1.37 ± 0.02 b	1.49 ± 0.01 a
Protein (g/100 g)	0.89 ± 0.03 c	1.17 ± 0.02 b	1.52 ± 0.03 a
Fat (g/100 g)	0.09 ± 0.01 c	0.17 ± 0.02 b	0.25 ± 0.01 a
Total carbohydrates (g/100 g)	6.21 ± 0.11 a	6.40 ± 0.15 a	6.47 ± 0.27 a
Total sugars (g/100 g)	4.07 ± 0.01 a	4.24 ± 0.10 a	4.39 ± 0.16 a
Salt (g/100 g)	1.04 ± 0.01 c	1.15 ± 0.01 b	1.27 ± 0.01 a
Energy value (kJ/100 g)	139.95 ± 0.52 c	154.63 ± 2.95 b	170.48 ± 3.28 a
TPC (mg GAE/g d.w.)	7.08 ± 0.41 a	7.00 ± 0.19 a	7.06 ± 0.22 a
DPPH IC_50_ (mg/mL)	267.97 ± 49.27 a	255.70 ± 32.24 a	171.09 ± 12.63 b
FRAP (mg AAE/g d.w.)	0.05 ± 0.00 c	0.42 ± 0.11 b	1.27 ± 0.23 a

Results are presented as mean ± standard deviation (*n* = 3). Results in a column marked with different letters are significantly different at the *p* < 0.05 level (Fischer’s LSD post hoc test).

**Table 3 foods-14-03539-t003:** Results of ANOVA for colour and texture measurement of juices.

Parameter	SC1	SC2	SC4	T1	T2	T4
CIE L*	24.54 ± 0.01 a	20.97 ± 0.01 b	20.60 ± 0.00 b	29.29 ± 0.01 a	21.94 ± 1.05 b	20.64 ± 0.01 c
CIE a*	15.21 ± 0.07 a	3.98 ± 0.03 b	3.65 ± 0.02 b	16.63 ± 0.05 a	6.09 ± 0.89 b	6.79 ± 0.07 b
CIE b*	7.24 ± 0.01 a	−0.21 ± 0.03 b	−1.86 ± 0.01 b	15.25 ± 0.01 a	1.84 ± 0.18 b	0.68 ± 0.02 c
CIE C*	16.85 ± 0.06 a	3.99 ± 0.03 c	4.10 ± 0.02 b	22.57 ± 0.04 a	6.36 ± 0.90 b	6.82 ± 0.07 b
CIE h°	25.45 ± 0.13 c	357.0 ± 0.3 a	332.9 ± 0.3 b	42.52 ± 0.08 a	16.97 ± 0.93 b	5.71 ± 0.19 c
Firmness (g)	16.59 ± 1.28 a	15.20 ± 0.14 a	15.97 ± 0.69 a	33.44 ± 0.93 a	24.90 ± 1.29 b	24.60 ± 1.21 b
Consistency (g s)	279.1 ± 2.3 b	277.8 ± 1.5 b	286.2 ± 4.5 a	529.6 ± 5.0 a	428.9 ± 9.5 b	439.1 ± 3.5 b
Cohesive-ness (g)	−13.85 ± 0.11 b	−14.25 ± 0.14 a	−14.19 ± 0.15 a	−12.82 ± 0.20 b	−16.33 ± 1.17 a	−17.55 ± 1.23 a
Index of viscosity (g s)	−1.30 ± 0.05 a	−1.31 ± 0.04 a	−1.32 ± 0.05 a	−7.44 ± 1.04 b	−7.43 ± 1.26 b	−10.06 ± 1.22 a

Results are presented as mean ± standard deviation (*n* = 8). Results in a row marked with different letters are significantly different at the *p* < 0.05 level (Fischer’s LSD post hoc test).

**Table 4 foods-14-03539-t004:** Results of ANOVA for descriptive analysis of juices.

Descriptor	SC1	SC2	SC4	T1	T2	T4
Blue colour intensity	0.3 ± 0.8 b	99.7 ± 0.8 a	99.3 ± 1.6 a	0.0 ± 0.0 b	99.7 ± 0.8 a	99.3 ± 1.6 a
Turbidity	98.0 ± 3.3 a	99.7 ± 0.8 a	99.7 ± 0.8 a	98.8 ± 2.0 a	99.8 ± 0.4 a	99.8 ± 4.0 a
Viscosity	44.3 ± 6.0 b	52.8 ± 7.1 ab	60.5 ± 15.6 a	50.8 ± 3.4 c	67.0 ± 8.3 b	80.5 ± 8.3 a
Granularity	12.8 ± 4.9 a	14.0 ± 10.8 a	16.5 ± 16.7 a	18.2 ± 6.7 b	27.7 ± 14.1 ab	39.0 ± 10.5 a
Characteristic odour	97.2 ± 4.5 a	46.0 ± 25.0 b	46.0 ± 21.5 b	100.0 ± 0.0 a	39.0 ± 8.9 b	30.7 ± 13.7 b
Algae odour	0.5 ± 1.2 b	18.0 ± 19.3 ab	28.7 ± 22.2 a	0.5 ± 1.2 b	41.0 ± 15.6 a	48.2 ± 24.4 a
Characteristic flavour	99.5 ± 1.2 a	70.8 ± 26.2 b	64.2 ± 23.8 b	99.2 ± 2.0 a	59.5 ± 18.7 b	54.2 ± 23.5 b
Algae flavour	0.2 ± 0.4 b	7.7 ± 8.3 b	22.5 ± 15.7 a	1.0 ± 1.5 c	13.0 ± 5.0 b	43.0 ± 15.3 a
Sweet taste	17.0 ± 11.2 a	14.2 ± 10.4 a	14.7 ± 8.2 a	22.7 ± 10.7 a	18.0 ± 5.9 a	14.7 ± 5.8 a
Salty taste	1.5 ± 2.5 a	0.5 ± 0.8 a	1.2 ± 2.9 a	64.2 ± 17.8 a	48.5 ± 14.4 ab	36.3 ± 10.5 b
Sour taste	64.3 ± 25.1 a	69.5 ± 30.9 a	72.2 ± 35.3 a	50.5 ± 20.5 b	68.7 ± 16.2 ab	74.8 ± 20.3 a
Bitter taste	0.0 ± 0.0 a	0.0 ± 0.0 a	0.5 ± 1.2 a	0.0 ± 0.0 b	0.0 ± 0.0 b	1.5 ± 1.8 a

Results are presented as mean ± standard deviation (*n* = 8). Results in a row marked with different letters are significantly different at the *p* < 0.05 level (Fischer’s LSD post hoc test).

**Table 5 foods-14-03539-t005:** Panel analysis.

	Factors			
Descriptors	*F*	*p*	*F*	*p*
Source	Products		Assessors	
Blue colour	24,663.069	<0.0001	0.172	0.970
Turbidity	1.086	0.392	4.951	0.003
Viscosity	26.235	<0.0001	7.307	0.000
Granularity	12.022	<0.0001	10.143	<0.0001
Characteristic odour	32.641	<0.0001	2.998	0.030
Algae odour	13.328	<0.0001	4.352	0.005
Mould odour	1.000	0.438	2.500	0.057
Characteristic flavour	12.692	<0.0001	6.468	0.001
Algae flavour	16.183	<0.0001	0.932	0.477
Mould flavour	1.000	0.438	24.391	<0.0001
Sweet taste	1.438	0.245	6.176	0.001
Salty taste	65.940	<0.0001	4.088	0.008
Sour taste	2.038	0.108	12.729	<0.0001
Bitter taste	2.946	0.032	1.161	0.356

*F*—*F*-values; *p*—*p*-values.

**Table 6 foods-14-03539-t006:** Distance to consensus computed across descriptors.

	Products					
Assessors	T1	T2	T4	SC1	SC2	SC4
A1	6.481	12.729	12.248	2.836	18.390	17.255
A2	8.984	14.227	13.593	2.843	7.658	9.956
A3	2.195	5.551	15.263	2.466	10.195	10.955
A4	2.749	6.449	9.824	3.627	9.145	11.767
A5	7.285	7.965	13.693	3.074	16.772	20.791
A6	4.914	12.311	13.511	2.270	14.217	13.762

## Data Availability

The original contributions presented in the study are included in the article/Appendix A, further inquiries can be directed to the corresponding author.

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
