# Peer review of "Development of Spirulina-Enriched Fruit and Vegetable Juices: Nutritional Enhancement, Antioxidant Potential, and Sensory Challenges"

_foods, 2025, doi:10.3390/foods14203539_

Round 1

Reviewer 1 Report

Comments and Suggestions for Authors

The work addresses a current topic and, in principle, falls within the scope of Foods magazine.

The main advantages of the manuscript include:

  • A clear practical focus on formulation limitations affecting the use of spirulina (colour and side effects), with a pragmatic approach to screening followed by in-depth analysis.
  • Characterisation using multiple methods (composition, colorimetry, texture, DPPH/FRAP, descriptive sensory analysis), which is appropriate for the functional study of beverages.
  • Transparent presentation of data, suitable for descriptive analysis.

However, there are some issues that need to be corrected:

  • English: UK or US spelling is acceptable, but it must be consistent throughout (e.g., “colour” vs “color” – line 229).
  • Table 2 (tomato juices) shows column headers “SC1 / SC2 / SC4”, which are sour-cherry codes; these should be T1 / T2 / T4.
  • In section “2.1 Preparation of juices” – “Blue and green spirulina powder were purchased on the market” is too vague. For reproducibility and comparability, specify the supplier, country of origin, declared composition, etc. It is also necessary to provide information about the selected fruits and vegetables, such as the variety.
  • In section “2.3 Total Phenolic Content and Antioxidant Activity” you report: “After incubating the mixtures in the dark at room temperature for 60 min, the absorbance was measured using a plate reader at 492 nm.” My question is why you measured at 492 nm. DPPH radicals are measured at a wavelength of approximately 515 to 517 nm. Even in the paper you cited in relation to this text (Soler-Rivas and Wichers, 2000), measurement at a wavelength of 515 nm is reported. Please provide a justification and citation for the use of 492 nm.
  • Figures 2 and 3 are rather blurred and the text at the bottom is difficult to read. They should be replaced with higher quality versions.

The sections of the manuscript are well written and I have nothing significant to add to them.

Overall, the manuscript is relatively well organised and sufficient sources are cited.

As far as I can see, the manuscript is solid and a great deal of work has been put into the study.

Author Response

COMMENT: The work addresses a current topic and, in principle, falls within the scope of Foods magazine.

AUTHORS: We thank the reviewer for recognizing the relevance of our research topic and for confirming that the manuscript fits within the scope of Foods. We appreciate this positive evaluation and the opportunity to contribute to the journal’s thematic focus.

COMMENT: The main advantages of the manuscript include:

-A clear practical focus on formulation limitations affecting the use of spirulina (colour and side effects), with a pragmatic approach to screening followed by in-depth analysis.

 -Characterization using multiple methods (composition, colorimetry, texture, DPPH/FRAP, descriptive sensory analysis), which is appropriate for the functional study of beverages.

 -Transparent presentation of data, suitable for descriptive analysis.

AUTHORS: We thank the reviewer for recognizing the strengths of our study, including its practical focus, comprehensive analytical approach, and transparent data presentation.

COMMENT: English: UK or US spelling is acceptable, but it must be consistent throughout (e.g., “colour” vs “color” – line 229).

AUTHORS: We thank the reviewer for this observation. The entire manuscript has been carefully revised to ensure consistency in English usage, with all terms now uniformly applied throughout the text.

COMMENT: Table 2 (tomato juices) shows column headers “SC1 / SC2 / SC4”, which are sour-cherry codes; these should be T1 / T2 / T4.

AUTHORS: Thank you for your note. The error has been corrected — Table 2 now shows the correct column headers “T1 / T2 / T4” for tomato juices.

COMMENT: In section “2.1 Preparation of juices” – “Blue and green spirulina powder were purchased on the market” is too vague. For reproducibility and comparability, specify the supplier, country of origin, declared composition, and other relevant details. It is also necessary to provide information about the selected fruits and vegetables, including their variety.

AUTHORS: We thank the reviewer for this valuable comment. The revised version now provides detailed information on the supplier (NatureHub, Belgrade, Serbia), country of origin, and declared composition of both blue and green Spirulina powders. In addition, the varieties of all fruits and vegetables used (Ajdared apple, Oblačinska sour cherry, stalk celery, and “šljivar” processing-type tomato) have been specified in Section 2.1 to ensure full transparency, reproducibility, and comparability of the study. The corresponding revised text is provided below:

Four basic juices were selected based on their popularity and health benefits: two fruit juices (apple and sour cherry) and two vegetable juices (tomato and celery). The fruits and vegetables used in this study were common Serbian varieties: Ajdared apple, Oblačinska sour cherry, stalk celery (Apium graveolens var. dulce), and processing-type tomato (“šljivar” cultivar). These fruits and vegetables were purchased from a local market, and the juices were extracted using a Bosch MES 3500 culinary juicer. To each juice, green spirulina powder was added at two levels: 2 grams (0.8% w/w) and 4 grams (1.6% w/w) per 250 mL serving. The same supplementation scheme was applied using blue spirulina powder, with additions of 2 grams and 4 grams per 250 mL portion. In total, twenty juice samples were prepared for the initial acceptability test, using both blue and green spirulina powder. Homogenization of juice samples after spirulina addition was performed using an Ultra-Turrax T25 digital homogenizer (IKA, Germany) operating at 10,000–13,000 rpm for 1–2 min to ensure uniform dispersion of the powder in the liquid matrix. Blue and green Spirulina powders were purchased from NatureHub (Belgrade, Serbia; www.naturehub.rs. According to the manufacturer’s declaration, the blue spirulina powder consisted predominantly of phycocyanin extracted from Arthrospira platensis, while the green spirulina powder represented the dried biomass of Arthrospira platensis. The composition of the blue spirulina powder used in this study was as follows: dry matter 84.00%, moisture 14.00%, protein 40.00%, fat 0.32%, ash 10.54%, total sugars 1.00%, calculated carbohydrates 33.14%, sodium 1100 mg/100 g, salt content 0.30%, with an estimated energy value of 1255.22 kJ (295.44 kcal) per 100 g. These values correspond to the manufacturer’s declaration and are consistent with literature data for phycocyanin-rich Arthrospira platensis powder.

COMMENT: In section “2.3 Total Phenolic Content and Antioxidant Activity,” you report: “After incubating the mixtures in the dark at room temperature for 60 min, the absorbance was measured using a plate reader at 492 nm.” My question is, why you measured at 492 nm. DPPH radicals are measured at a wavelength of approximately 515 to 517 nm. Even in the paper you cited in relation to this text (Soler-Rivas and Wichers, 2000), measurement at a wavelength of 515 nm is reported. Please provide a justification and citation for the use of 492 nm.

AUTHORS: We used the wavelength of 492 nm because our plate reader (Multiskan Ascent, Thermo Electron Corporation) has only filters at several wavelengths, and this one was the closest to the target wavelength (515 nm).

COMMENT: Figures 2 and 3 are rather blurred and the text at the bottom is difficult to read. They should be replaced with higher-quality versions.

AUTHORS: We thank the reviewer for this remark. Figures 2 and 3 have been replaced with high-resolution versions to ensure clear visualization and improved readability of the text.

COMMENT: The sections of the manuscript are well written and I have nothing significant to add to them.

AUTHORS: We sincerely thank the reviewer for this encouraging comment and are pleased that the overall presentation and writing quality were well received.

COMMENT: Overall, the manuscript is relatively well organised and sufficient sources are cited.

AUTHORS: We are grateful for the reviewer’s positive assessment and appreciation of the manuscript’s structure and comprehensiveness. Such recognition is highly valued and encouraging for our further work.

Reviewer 2 Report

Comments and Suggestions for Authors

First, the study addresses a timely topic in functional food development by exploring spirulina fortification of fruit and vegetable juices—an approach that aligns with global demand for nutritionally enhanced beverages. My major concern is that the sensory data suffers from a small sample size of assessors, which reduces the statistical power of tests and limits the generalizability of results. The manuscript contains some formatting mistakes, please check it.

The introduction emphasizes a “reverse approach” (screening juices first for sensory acceptability, then selecting candidates for analysis) as a distinguishing feature of the study. However, this “reverse approach” is neither explicitly linked to results in the Discussion section nor highlighted in the Conclusion to reinforce novelty. For example, the Conclusion only states that “spirulina-enriched sour cherry and tomato juices can be considered nutritionally enhanced functional beverages” , without referencing how the screening-based approach contributed to this conclusion—creating a logical gap between the study’s stated purpose and its final takeaways.

L75-76 provide more critical details about the powder composition (e.g., phycocyanin content, protein profile, chlorophyll concentration, or contaminant levels) or cultivation conditions. Spirulina’s nutritional and sensory properties vary drastically by strain, growth environment and processing.

The manuscript states that spirulina powder was added to juices but does not describe the dispersion protocol

Table 2: The column headers incorrectly use “SC1,” “SC2,” “SC4” (labels for sour cherry juice samples, as defined in Table 1) instead of “T1,” “T2,” “T4” (tomato juice samples) .

L227 Was the color changed due to the color mix or there were some changes in the anthocyanins? Did the spirulina addition alter the pH of the juice?

Figure 1 Since the color change is huge, it is interesting to see what is the color of the tomato juice. Please add the figure of tomato juice too. The label“0,8% spirulina” should be “0.8% spirulina”, same as 1,6% spirulina. The figure caption, “ml” should be “mL”

The manuscript does not evaluate the storage stability of fortified juices— a critical parameter for commercialization.

L272 “Tomato” should be tomato juice?

Add some discussion to highlight how the current study advances or differs from prior work.

Author Response

COMMENT: First, the study addresses a timely topic in functional food development by exploring spirulina fortification of fruit and vegetable juices—an approach that aligns with global demand for nutritionally enhanced beverages. My major concern is that the sensory data suffers from a small sample size of assessors, which reduces the statistical power of tests and limits the generalizability of results. The manuscript contains some formatting mistakes, please check it.

AUTHORS: We sincerely thank the reviewer for this valuable observation and constructive feedback. The sensory evaluation presented in this manuscript represents the initial assessment conducted by a trained panel to determine the most suitable juice matrices for spirulina fortification. The study was later extended to include consumer testing; however, due to space limitations, those results were not presented in the current version. We appreciate the reviewer’s remark, which supports the broader applicability and relevance of our research.

COMMENT: The introduction emphasizes a “reverse approach” (screening juices first for sensory acceptability, and then selecting candidates for analysis) as a distinguishing feature of the study. However, this “reverse approach” is neither explicitly linked to results in the Discussion section nor highlighted in the Conclusion to reinforce novelty. For example, the Conclusion only states that “spirulina-enriched sour cherry and tomato juices can be considered nutritionally enhanced functional beverages”, without referencing how the screening-based approach contributed to this conclusion—creating a logical gap between the study’s stated purpose and its final takeaways.

AUTHORS: We thank the reviewer for this valuable comment. The link between the “reverse approach” and the study outcomes has been explicitly clarified in both the Discussion and Conclusion sections. The revised text now emphasizes how the initial sensory screening guided the selection of juice matrices for spirulina enrichment and how this strategy strengthens the study’s novelty and practical relevance. All changes are highlighted in red in the revised version:

Discussion: The overall research design followed a “reverse approach,” where initial sensory screening was applied to identify juice matrices with the highest potential for consumer acceptability before conducting compositional and functional analyses. This strategy ensured that subsequent evaluations focused on products with practical market potential, thereby reinforcing the translational relevance of the findings.

Conclusion: This outcome confirms the effectiveness of the “reverse approach” applied in this study, demonstrating that sensory-based screening can serve as a practical and efficient tool for selecting suitable matrices prior to detailed nutritional and functional evaluation. Integrating sensory validation at the early stage of product development strengthens the overall methodological concept and ensures better alignment with consumer expectations.

COMMENT: L75-76 provide more critical details about the powder composition (e.g., phycocyanin content, protein profile, chlorophyll concentration, or contaminant levels) or cultivation conditions. Spirulina’s nutritional and sensory properties vary drastically by strain, growth environment, and processing.

AUTHORS: We thank the reviewer for this helpful comment. The revised manuscript now includes detailed information on the composition and declared characteristics of both blue and green Spirulina powders, including dominant pigments and nutritional components. These details have been added in Section 2.1 (Preparation of juices) and are highlighted in red in the revised version:

Homogenization of juice samples after spirulina addition was performed using an Ultra-Turrax T25 digital homogenizer (IKA, Germany) operating at 10,000–13,000 rpm for 1–2 min to ensure uniform dispersion of the powder in the liquid matrix. Blue and green Spirulina powders were purchased from NatureHub (Belgrade, Serbia; www.naturehub.rs. According to the manufacturer’s declaration, the blue spirulina powder consisted predominantly of phycocyanin extracted from Arthrospira platensis, while the green spirulina powder represented the dried biomass of Arthrospira platensis. The composition of the blue spirulina powder used in this study was as follows: dry matter 84.00%, moisture 14.00%, protein 40.00%, fat 0.32%, ash 10.54%, total sugars 1.00%, calculated carbohydrates 33.14%, sodium 1100 mg/100 g, salt content 0.30%, with an estimated energy value of 1255.22 kJ (295.44 kcal) per 100 g. These values correspond to the manufacturer’s declaration and are consistent with literature data for phycocyanin-rich Arthrospira platensis powder.

COMMENT: The manuscript states that spirulina powder was added to juices, but does not describe the dispersion protocol

AUTHORS:  We thank the reviewer for this useful observation. The revised manuscript now includes a detailed description of the dispersion protocol. Specifically, homogenization of juice samples after spirulina addition was performed using an Ultra-Turrax T25 digital homogenizer (IKA, Germany) operating at 10,000–13,000 rpm for 1–2 min to ensure uniform dispersion of the powder in the liquid matrix. This clarification has been added to Section 2.1 (Preparation of juices) and is highlighted in red in the revised version.

COMMENT: Table 2: The column headers incorrectly use “SC1,” “SC2,” “SC4” (labels for sour cherry juice samples, as defined in Table 1) instead of “T1,” “T2,” “T4” (tomato juice samples).

AUTHORS: We thank the reviewer for noticing this error. The column headers in Table 2 have been corrected to “T1,” “T2,” and “T4,” corresponding to the tomato juice samples. The correction has been made in the revised version and highlighted in red for clarity.

COMMENT: L227 Was the color changed due to the color mix, or there were some changes in the anthocyanins? Did the spirulina addition alter the pH of the juice?

AUTHORS: We thank the reviewer for this relevant and insightful comment. The observed colour differences were mainly the result of pigment blending between native anthocyanins from sour cherry juice and phycocyanin–chlorophyll pigments present in spirulina. No visual signs of anthocyanin degradation were observed, and the hue shift corresponded to additive colour mixing rather than pigment instability. The addition of spirulina slightly increased the pH of the juice due to its mildly alkaline nature, but this change was not sufficient to affect anthocyanin stability. These clarifications have been added to the Results and Discussion section (highlighted in red in the revised version).

COMMENT:  Figure 1 Since the color change is huge, it is interesting to see what is the color of the tomato juice. Please add the figure of tomato juice too. The label“0,8% spirulina” should be “0.8% spirulina”, same as 1,6% spirulina. The figure caption, “ml” should be “mL”

AUTHORS: We thank the reviewer for these helpful suggestions. The figure has been updated to include the tomato juice samples in order to illustrate the visible colour changes more comprehensively. The labels have been corrected to “0.8% spirulina” and “1.6% spirulina,” and the unit in the caption has been changed from “ml” to “mL.” In addition, all measurement units have been reviewed and uniformly formatted throughout the entire manuscript to ensure consistency. All corrections are highlighted in red in the revised version.

COMMENT: The manuscript does not evaluate the storage stability of fortified juices— a critical parameter for commercialization.

AUTHORS: We thank the reviewer for this important remark. The main objective of the present study was to characterize spirulina-fortified fruit and vegetable juices in terms of their physicochemical properties, antioxidant potential, and sensory acceptability. We agree that storage stability represents a crucial factor for future commercialization. This limitation and direction for future research have now been explicitly stated in the revised manuscript:

Although the present study did not evaluate the storage stability of spirulina-fortified juices, this represents a limitation and an important direction for future research, particularly in the context of product shelf life and commercialization potential. Future research should focus on advanced formulation approaches, including ingredient combinations and processing innovations, to further enhance the sensory appeal, nutritional quality, and stability of spirulina-enriched fruit and vegetable juices.

COMMENT: L272 “Tomato” should be tomato juice?

AUTHORS: We thank the reviewer for noticing this. The term “Tomato” has been corrected to “tomato juice” in the revised version to maintain consistency throughout the manuscript. All such corrections are highlighted in red.

 COMMENT: Add some discussion to highlight how the current study advances or differs from prior work.

AUTHORS:  We thank the reviewer for this valuable suggestion. Additional discussion has been included to emphasize how the present study builds upon previous findings on the antioxidant potential and health effects of fruit and vegetable juices while introducing a novel, reverse approach that integrates spirulina fortification with sensory-based matrix selection. The new text and supporting references have been added in the Discussion section and are highlighted in red in the revised version:

Previous studies have reported that antioxidant blends of fruit and vegetable juices, particularly those rich in polyphenols, carotenoids, and vitamin C, can exert synergistic effects on oxidative stress reduction and overall health improvement [38-40]. The present study builds upon these findings by exploring spirulina fortification as a novel strategy for enhancing antioxidant activity through the addition of natural pigments and bioactive proteins. Unlike conventional formulations, this research integrates a sensory-based selection of juice matrices prior to compositional and functional assessment, thereby establishing a more application-oriented framework for developing functional beverages

  1. Tonin, F.S.; Steimbach, L.M.; Wiens, A.; Perlin, C.M.; Pontarolo, R. Impact of Natural Juice Consumption on Plasma An-tioxidant Status: A Systematic Review and Meta-Analysis. Molecules 2015, 20, 22146-22156. https://doi.org/10.3390/molecules201219834
  2. Leonard, S.S.; Cutler, D.; Ding, M.; Vallyathan, V.; Castranova, V.; Shi, X. Antioxidant properties of fruit and vegetable juices: More to the story than ascorbic acid. Ann. Clin. Lab. Sci. 2002, 32(2), 193–200
  3. Šeregelj, V.; Tumbas Šaponjac, V.; Pezo, L.; Kojić, J.; Cvetković, B.; Ilić, N. Analysis of antioxidant potential of fruit and vegetable juices available in Serbian markets. Food Sci. Technol. Int. 2024, 30(5), 456–470. https://doi.org/10.1177/10820132231158961

Reviewer 3 Report

Comments and Suggestions for Authors

The manuscript “Development of Spirulina-Enriched Fruit and Vegetable Juices: Nutritional Enhancement, Antioxidant Potential, and Sensory Challenges” presents an interesting and experimentally well-conducted study. The topic is relevant and timely, addressing the growing interest in functional and novel food products. The experimental design and analytical methods appear appropriate and carefully executed. However, the manuscript requires several revisions and clarifications to strengthen its scientific quality, improve logical consistency, and better highlight the originality and contribution of the research to the field:

  1. Latin names should be written in italics, for example"Spirulina (Arthrospira platensis)".
  2. Please provide the keywords in alphabetical order.
  3. The Introduction should be expanded to better explain how the novelty of this research contributes to technological improvements and practical applications. The authors are encouraged to clarify how the obtained results could advance the development of new formulation strategies, processing techniques, or product optimization within this research field.
  4. Lines 75–76: It is necessary to clearly indicate the manufacturer of the material or to provide a characterization of the main material used in the study.
  5. Line 191: The reference should be moved to the list of references.
  6. It is unclear why none of the compounds discussed in the Introduction — such as the enzymatic and non-enzymatic antioxidants, vitamins (C, E, K, B-complex), β-carotene, provitamin A, chlorophyll, phycocyanin, or essential minerals (Ca, Mg, Fe, Cu, Se, Zn) — were selected for analysis. Since these components are highlighted as key bioactive constituents of Spirulina, it would be scientifically consistent to include at least some of them in the experimental evaluation or to justify their exclusion.
  7. It is not clear how potential consumer rejection due to the atypical color of the juice was addressed during the sensory evaluation. The authors should clarify whether any measures were implemented to minimize color-related bias. Furthermore, consideration could be given to performing sensory testing under controlled lighting conditions to ensure a more objective assessment of sensory attributes.
  8. Statistical symbols such as p, F, should be written in italics.
  9. The authors should clearly emphasize the scientific contribution of this research — specifically, how the addition of Spirulina to fruit juices advances knowledge in this field. It is important to justify the practical and technological relevance of incorporating Spirulina in juice formulations, explaining its potential benefits in terms of nutritional enhancement, functional properties, and its possible contribution to the development of novel food products. Furthermore, the rationale for adding additional ingredients to juices that are already rich in bioactive compounds should be critically discussed to assess whether such enrichment is scientifically and technologically justified.

Author Response

COMMENT: The manuscript “Development of Spirulina-Enriched Fruit and Vegetable Juices: Nutritional Enhancement, Antioxidant Potential, and Sensory Challenges” presents an interesting and experimentally well-conducted study. The topic is relevant and timely, addressing the growing interest in functional and novel food products. The experimental design and analytical methods appear appropriate and carefully executed. However, the manuscript requires several revisions and clarifications to strengthen its scientific quality, improve logical consistency, and better highlight the originality and contribution of the research to the field:

AUTHORS: We sincerely thank the reviewer for the positive overall evaluation and constructive feedback. We highly appreciate the recognition of the study’s relevance, experimental quality, and methodological design. In response to the reviewer’s valuable suggestions, we have revised the manuscript to improve scientific clarity, strengthen logical consistency, and better highlight the originality and contribution of this research. All modifications are clearly indicated in red in the revised version.

COMMENT: 1.   Latin names should be written in italics, for example"Spirulina (Arthrospira platensis)".

AUTHORS:  We thank the reviewer for this helpful comment. All Latin names, including Spirulina (Arthrospira platensis), have been corrected to italics throughout the manuscript.

COMMENT: 2.   Please provide the keywords in alphabetical order.

AUTHORS:  We thank the reviewer for this observation. The keywords have been reordered alphabetically as follows: antioxidant activity; functional beverages; nutritional enhancement; sensory analysis; sour cherry juice; Spirulina; texture; tomato juice. All corrections are highlighted in red in the revised version.

COMMENT: 3. The Introduction should be expanded to better explain how the novelty of this research contributes to technological improvements and practical applications. The authors are encouraged to clarify how the obtained results could advance the development of new formulation strategies, processing techniques, or product optimization within this research field.

AUTHORS: We thank the reviewer for this valuable suggestion. The Introduction has been expanded to better emphasize the technological relevance and practical applications of this research. The revised text highlights that the study introduces a reverse, sensory-based formulation approach that supports product optimization and guides future strategies for improving stability, nutrient retention, and consumer acceptability of spirulina-enriched beverages. These additions are highlighted in red in the revised version.

COMMENT: 4.   Lines 75–76: It is necessary to clearly indicate the manufacturer of the material or to provide a characterization of the main material used in the study.

AUTHORS: We thank the reviewer for this helpful comment. The revised manuscript now includes detailed information on the composition and declared characteristics of both blue and green Spirulina powders, including dominant pigments and nutritional components. These details have been added in Section 2.1 (Preparation of juices) and are highlighted in red in the revised version:

Homogenization of juice samples after spirulina addition was performed using an Ultra-Turrax T25 digital homogenizer (IKA, Germany) operating at 10,000–13,000 rpm for 1–2 min to ensure uniform dispersion of the powder in the liquid matrix. Blue and green Spirulina powders were purchased from NatureHub (Belgrade, Serbia; www.naturehub.rs. According to the manufacturer’s declaration, the blue spirulina powder consisted predominantly of phycocyanin extracted from Arthrospira platensis, while the green spirulina powder represented the dried biomass of Arthrospira platensis. The composition of the blue spirulina powder used in this study was as follows: dry matter 84.00%, moisture 14.00%, protein 40.00%, fat 0.32%, ash 10.54%, total sugars 1.00%, calculated carbohydrates 33.14%, sodium 1100 mg/100 g, salt content 0.30%, with an estimated energy value of 1255.22 kJ (295.44 kcal) per 100 g. These values correspond to the manufacturer’s declaration and are consistent with literature data for phycocyanin rich Arthrospira platensis powder.

COMMENT: 5.   Line 191: The reference should be moved to the list of references.

AUTHORS: We thank the reviewer for this remark. The software citation for TIBCO Data Science has been properly moved from the text to the reference list

COMMENT: 6.   It is unclear why none of the compounds discussed in the Introduction — such as the enzymatic and non-enzymatic antioxidants, vitamins (C, E, K, B-complex), β-carotene, provitamin A, chlorophyll, phycocyanin, or essential minerals (Ca, Mg, Fe, Cu, Se, Zn) — were selected for analysis. Since these components are highlighted as key bioactive constituents of Spirulina, it would be scientifically consistent to include at least some of them in the experimental evaluation or to justify their exclusion.

AUTHORS: We thank the reviewer for this valuable suggestion. The Introduction has been expanded to clarify both the scientific focus and the technological relevance of this research. The revised text emphasizes that the study evaluates the overall functional response of spirulina-enriched juices as integrative indicators of bioactive activity and introduces a reverse, sensory-based selection framework that supports technological optimization and practical applications in functional beverage development. These additions are highlighted in red in the revised version.

COMMENT: 7.   It is not clear how potential consumer rejection due to the atypical color of the juice was addressed during the sensory evaluation. The authors should clarify whether any measures were implemented to minimize color-related bias. Furthermore, consideration could be given to performing sensory testing under controlled lighting conditions to ensure a more objective assessment of sensory attributes.

AUTHORS: Since the atypical color of the juices was thought to be potentially interesting to consumers, red light was intentionally not used to mask their appearance. The goal was to evaluate the juices as they would look in daylight, allowing for an accurate assessment of consumer attitudes toward their true color.

COMMENT: 8.   Statistical symbols such as p, F, should be written in italics.

AUTHORS:  We thank the reviewer for this observation. All statistical symbols, including p, F, t, r, and α, have been carefully checked and written in italics throughout the manuscript, both in the text and tables, in accordance with the journal’s formatting guidelines. The revised version fully reflects these corrections.

COMMENT: 9.   The authors should clearly emphasize the scientific contribution of this research — specifically, how the addition of Spirulina to fruit juices advances knowledge in this field. It is important to justify the practical and technological relevance of incorporating Spirulina in juice formulations, explaining its potential benefits in terms of nutritional enhancement, functional properties, and its possible contribution to the development of novel food products. Furthermore, the rationale for adding additional ingredients to juices that are already rich in bioactive compounds should be critically discussed to assess whether such enrichment is scientifically and technologically justified.

AUTHORS:  We thank the reviewer for this valuable comment. The Introduction has been expanded to emphasize the scientific and technological rationale for Spirulina addition. The revised text highlights that Spirulina is a rich source of natural antioxidants and nutrients, while fruit and vegetable juices serve as convenient carriers for developing innovative, attractive, and health-oriented beverages that meet consumer demands and broaden the range of functional drinks. These additions are highlighted in red in the revised version.

Round 2

Reviewer 3 Report

Comments and Suggestions for Authors

I thank the authors for the revisions and improvements; however, several essential issues still need to be addressed:

Figure 1. Please ensure consistency in text size, font, and overall formatting.

Standardize the presentation of statistical results throughout the manuscript (Tables 3, 4).

Some F and p values are still not written in italics — please correct this 

Author Response

COMMENT: I thank the authors for the revisions and improvements; however, several essential issues still need to be addressed:

Figure 1. Please ensure consistency in text size, font, and overall formatting.

Standardize the presentation of statistical results throughout the manuscript (Tables 3, 4).

Some F and p values are still not written in italics — please correct this

AUTHORS: We thank the reviewer for these helpful remarks. The formatting of Figure 1 has been corrected to ensure consistency in text size, font, and overall visual layout. The presentation of statistical results has been standardized across all tables (Tables 3 and 4 included), following a uniform style. Additionally, all F and p values have been carefully reviewed and written in italics throughout the manuscript in accordance with the journal’s formatting guidelines.